# Some Qualitative Behavior of Solutions of General Class of Difference Equations

**Osama Moaaz [1], Dimplekumar Chalishajar [2,\*] and Omar Bazighifan [3]** 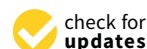

[1]  Department of Mathematics, Faculty of Science, Mansoura University, Mansoura 35516, Egypt
[2]  Department of Applied Mathematics, Virginia Military Institute (VMI) 435 Mallory Hall, Lexington, VA 24450, USA
[3]  Department of Mathematics, Faculty of Science, Hadhramout University, Hadhramout 50512, Yemen
\*   Correspondence: chalishajardn@vmi.edu

**Abstract:** In this work, we consider the general class of difference equations (covered many equations that have been studied by other authors or that have never been studied before), as a means of establishing general theorems, for the asymptotic behavior of its solutions. Namely, we state new necessary and sufficient conditions for local asymptotic stability of these equations. In addition, we study the periodic solution with period two and three. Our results essentially extend and improve the earlier ones.

**Keywords:** qualitative behavior; general class of difference equations; locally asymptotically stable; period two and three

## 1. Introduction

Difference equations are recognized as description of observed evolution of a phenomenon, where the majority of measurements of a time-evolving variable are discrete. As a result, these equations get their importance in arithmetical models. Moreover, difference equations are used in studying of discrete approaches. Many results have been obtained in the theory of difference equations as more natural discrete analogues of corresponding results of differential equations. However, this could be true particularly in case of the stability theory of Lyapunov. Moreover, the applications of the differential equation theory are growing rapidly in a wide variety of fields (i.e., numerical analyses, finite mathematics, control theory, and computer science). This puts lots of potential on studying the difference equations theory in addition to the related disciplines. Difference equations have been widely used as mathematical models for describing real life situations in probability theory, queuing problems, stochastic time series, combinatorial analysis, statistical problems, number theory, geometry, electrical networks, quanta in radiation, economics, genetics in biology, psychology, sociology, refer [1–8]. For varied reasons, rational difference equations have sparked the debate of many researchers. First, they afford many examples of non-linear equations which are treatable, in many cases. However, their dynamics offer some strong features with regard to the linear case. In fact, the importance of studying difference equations comes from their appearance in many biological models which have many applications. One of the interesting examples for both facts is Riccati difference equations; the richness of the dynamics of Riccati equations is very well-known [9], and a specific case of these equations provides the classical Beverton-Holt model on the dynamics of exploited fish populations [10]. Clearly, higher-order rational difference equations and systems of rational equations

have also been widely studied but yet leave many aspects to be investigated. For many results, applications, and open problems on higher-order equations and difference systems, refer [8,11].

Our aim in this paper is to investigate the qualitative behavior of solutions of the general equation

$$J_{n+1} = f(J_n, J_{n-1}), \quad n = 0, 1, \ldots \tag{1}$$

where the function $f$ is continuous real function and homogenous with degree *wero* and the initial condition $J_{-1}$ and $J_0$ are real numbers.

There are many papers devoted to the qualitative behavior of solutions of special cases of Equation (1) (see [12–17]). Most of these papers studied only the sufficient condition for stability of equilibrium point. Our aim here is to give a complete picture regarding the stability of equilibrium point of general Equation (1). In other words, we state a new necessary and sufficient condition for locally asymptotically stability of Equation (1) which extends and complements the earlier ones. Moreover, by using a new technique, we study the existence of periodic solutions of prime period two and three. New periodicity conditions are extended to a number of existing conditions.

Let difference equation

$$J_{n+1} = F(J_n, J_{n-1}, \ldots, J_{n-k}), \quad n = 0, 1, \ldots \tag{2}$$

has positive equilibrium point $J^*$, the linearized equation of Equation (2) of $J^*$ is

$$J_{n+1} + \eta_0 J_n + \eta_1 J_{n-1} + \cdots + \eta_k J_{n-k} = 0, \tag{3}$$

where $\eta_\zeta = -F_{u_\zeta}(J^*, J^*, \ldots, J^*)$ for $\zeta = 0, 1, \ldots, k$. Then, the characteristic equation of Equation (3) is

$$\gamma^{k+1} + \eta_0 \gamma^k + \eta_1 \gamma^{k-1} + \cdots + \eta_k = 0. \tag{4}$$

Equation (3) will be called stable, asymptotically stable, or unstable provided that the zero equilibrium point has that property. In addition, the asymptotic stability of the zero equilibrium is equivalent to every solution having limit zero as $n \to \infty$ which in turn is true if and only if all roots of the characteristic equation lie in the open unit disk $|\gamma| < 1$. A positive semicycle of a solution $\{J_n\}_{n=-1}^\infty$ of (1) consists of a "string" of terms $\{J_r, J_{r+1}, \ldots, J_s\}$, all greater than or equal to the equilibrium $J^*$, with $r \geq -1$ and $s \leq \infty$ and such that

$$\text{either } r = -1 \text{ or } r > -1 \text{ and } J_{r-1} < J^*$$

and

$$\text{either } m = \infty \text{ or } m < \infty \text{ and } J_{m+1} < J^*.$$

A negative semicycle of a solution $\{J_n\}_{n=-1}^\infty$ of (1) consists of a "string" of terms $\{J_r, J_{r+1}, \ldots, J_s\}$, all less than the equilibrium $J^*$, with $r \geq -1$ and $s \leq \infty$ and such that

$$\text{either } r = -1 \text{ or } r > -1 \text{ and } J_{r-1} \geq J^*$$

and

$$\text{either } m = \infty \text{ or } m < \infty \text{ and } J_{m+1} \geq J^*.$$

## 2. Main Results

*2.1. Local Stability and Semi-Cycle Analysis*

Next, we study the asymptotic stability for (1). The equilibrium point of Equation (1) is given by

$$J^* = f(1,1).$$

The linearized equation associated with (1) about $J^*$ is

$$\omega_{n+1} - f_u(J^*, J^*)\,\omega_n - f_v(J^*, J^*)\,\omega_{n-1} = 0. \tag{5}$$

From Euler's homogeneous function Theorem, we have that $u f_u + v f_v = 0$, and hence $f_u(J^*, J^*) = -f_v(J^*, J^*)$. In addition, from [18], we get that $f_u$ and $f_v$ homogenous with degree $-1$ and hence (5) yield

$$\omega_{n+1} - \mu\omega_n + \mu\omega_{n-1} = 0, \tag{6}$$

where

$$\mu = \frac{f_u(1,1)}{f(1,1)}.$$

In the next theorems, we study the asymptotic stability for (1).

**Theorem 1.** *For local stability of the equilibrium point $J^*$ of Equation (1), we have the following cases:*

1. *If $\mu \in (0,1)$, then $J^*$ is locally asymptotically stable and sink.*
2. *If $\mu > 1$, then $J^*$ is unstable and repeller.*
3. *If $\mu \in (-1/2, 0)$, then $J^*$ is locally asymptotically stable and sink.*
4. *If $\mu < -1/2$, then $J^*$ is an unstable saddle point.*
5. *If $\mu = 1$ or $-1/2$, then $J^*$ is a nonhyperbolic point.*

**Proof.** The characteristic equation of (6) is

$$\gamma^2 - \mu\gamma + \mu = 0. \tag{7}$$

Assume that $\mu > 0$. From [19], we have that $J^*$ is locally asymptotically stable if and only if $\mu < 1$ and $J^*$ is sink. Otherwise, if $\mu > 1$, then $J^*$ is unstable,

$$|\mu| > 1 \text{ and } |\mu| < |1 + \mu|,$$

and hence, from Theorem 1.1.1-(d) in [20], $J^*$ is a repeller.

On the other hand, assume that $\mu < 0$. Let $\mu = -\delta < 0$, then the roots of the characteristic Equation (7) are

$$\gamma_\pm = \frac{\delta}{2}\left(-1 \pm \sqrt{1 + \frac{4}{\delta}}\right).$$

We define the following functions

$$g(s) = \frac{s}{2}\left(-1 + \sqrt{1 + \frac{4}{s}}\right), \tag{8}$$

$$h(s) = \frac{s}{2}\left(-1 - \sqrt{1 + \frac{4}{s}}\right). \tag{9}$$

Now, let $\delta > \frac{1}{2}$. From definition of $h(s)$, we have $h'(s) < 0$ for $s > 0$. Since $h(s)$ decreasing and $h(1/2) = -1$, we get $h(s) < -1$ for $s > 1/2$ and hence $|\gamma_-| > 1$. Thus, $J^*$ is unstable if $\delta > \frac{1}{2}$ $\left( \Longleftrightarrow \mu < -\frac{1}{2} \right)$. In addition, we note that

$$\mu^2 - 4\mu > \frac{1}{4} + 4\left(\frac{1}{2}\right) = \frac{9}{4} > 0 \ \text{ and } \ |\mu| > |1 + \mu| \ \text{ for all } \mu < -\frac{1}{2},$$

Hence, from Theorem 1.1.1-(e) in [20], $J^*$ is an unstable saddle point.

Next, let $0 < \delta < 1/2$. From (8), we get

$$g'(s) = \frac{1}{2}\left(-1 + \frac{s+2}{\sqrt{s^2 + 4s}}\right).$$

Since $(s+2)^2 > (s^2 + 4s)$, we obtain $\frac{s+2}{\sqrt{s^2+4s}} > 1$ and hence $g'(s) > 0$. Therefore, $0 < g(s) < 1/2$ for $s \in (0, 1/2)$ and then $|\gamma_+| < 1$. In addition, from (9), we see that $h(s)$ decreasing for $s \in (0, 1/2)$ and $-1 < h(s) < 0$, then $|\gamma_-| < 1$. Thus, we obtain $|\gamma_\pm| < 1$, and hence $J^*$ is locally asymptotically stable and sink if $0 < \delta < 1/2$.

Finally, if $\mu = 1$ or $-1/2$, then

$$\begin{aligned} |\mu| &= |1 + \mu|, & \text{if } \mu = -1/2, \\ -\mu &= -1 \text{ and } |\mu| < 2 & \text{if } \mu = 1, \end{aligned}$$

Thus, from Theorem 1.1.1-(e) in [20], $J^*$ is a nonhyperbolic point. Then, the proof is completed. $\quad\square$

**Remark 1.** *From Theorem 1, we see that the equilibrium point $J^*$ of Equation (1) is locally asymptotically stable if*

$$0 < \mu < 1 \quad or \quad -\frac{1}{2} < \mu < 0.$$

*and unstable if*

$$\mu > 1 \quad or \quad \mu < -\frac{1}{2}.$$

Next, we give some results about the semi-cycles of Equation (1).

**Theorem 2.** *Let $f \in C\left((0, \infty) \times (0, \infty), (0, \infty)\right)$ and $J^*$ be a positive equilibrium of Equation (1). For a semi-cycle analysis of the solutions of Equation (1), we have only the following two cases:*
*($c_1$) Except possibly for the first semi-cycle, every solution of Equation (1) has semi-cycles of length one.*
*($c_2$) Except possibly for the first semicycle, every oscillatory solution of Equation (1) has semi-cycles of length at least two.*

**Proof.** Since the function $f : (0, \infty)^2 \to (0, \infty)$ homogenous with degree *wero*, we get from Euler's homogeneous function Theorem

$$\frac{f_u}{f_v} = -\frac{v}{u} < 0$$

Hence, we have two cases

$$\textbf{(i) } f_u < 0 \ \text{ and } \ f_v > 0 \tag{10}$$

or

$$\textbf{(ii) } f_u > 0 \ \text{ and } \ f_v < 0. \tag{11}$$

For case (**i**), if $\{J_n\}_{n=-1}^{\infty}$ is a solution of Equation (1) with at least two semi-cycles, then there exists a positive integer $m$ such that either $J_{m-1} < J^* \leq J_m$ or $J_{m-1} \geq J^* > J_m$. Let $J_{m-1} < J^* \leq J_m$. From (10), we have

$$
\begin{aligned}
J_{m+1} &= f(J_m, J_{m-1}) \\
&< f(J^*, J_{m-1}) \\
&< f(J^*, J^*) \\
&= f(1, 1) = J^*
\end{aligned}
$$

and

$$
\begin{aligned}
J_{m+2} &= f(J_{m+1}, J_m) \\
&> f(J^*, J^*) = J^*.
\end{aligned}
$$

Thus, it is followed by the induction that

$$
J_{m+2s-1} < J^* < J_{m+2s} \ \text{ for all } s = 0, 1, 2, \dots .
$$

In the case where $J_{m-1} \geq J^* > J_m$, the proof is similar and is omitted. Then $\{J_n\}_{n=-1}^{\infty}$ has semicycles of length one.

For case (**ii**), if $\{J_n\}_{n=-1}^{\infty}$ is an oscillatory solution with $J_{m-1} < J^* \leq J_m$, then we get

$$
\begin{aligned}
J_{m+1} &= f(J_m, J_{m-1}) \\
&> f(J^*, J^*) = J^*.
\end{aligned}
$$

Thus, the term $J_{m+1}$ also belongs to the positive semicycle and hence $\{J_n\}_{n=-1}^{\infty}$ has semi-cycles of length at least two. Similarly, we can prove the case where $J_{m-1} \geq J^* > J_m$ which is omitted here for convenience. Then, the proof is completed. $\square$

**Theorem 3.** *Let $f \in C\left((0, \infty) \times (0, \infty), (0, \infty)\right)$, $J^*$ be a positive equilibrium of Equation (1), $f_u > 0$ and $f_v < 0$. If*

$$
f(u, v) < u \ \text{ for all } v \in (J^*, u), \tag{12}
$$

*then, except possibly for the first semicycle which may have length one, every oscillatory solution of Equation (1) has semi-cycles of length two or three. Whereas if*

$$
f(u, v) > u \ \text{ for all } v \in (J^*, u), \tag{13}
$$

*then, every solution of Equation (1) with $J_0 > J_{-1} > J^*$ is increasing.*

**Proof.** Assume that $\{J_n\}_{n=-1}^{\infty}$ is an oscillatory solution of Equation (1). From Theorem 2, we get that $\{J_n\}_{n=-1}^{\infty}$ has semi-cycles of length at least two. Then, there exists a positive integer $m$ such that either

$$
J_{m-1} < J^* < J_{m+1} < J_m \tag{14}
$$

or

$$
J_{m-1} < J^* < J_m < J_{m+1} \tag{15}
$$

Let (12) hold. Now, if we have (6) holds, then we get

$$
\begin{aligned}
J_{m+2} &= f(J_{m+1}, J_m) \\
&< f(J_m, J_m) = f(1, 1) = J^*,
\end{aligned}
$$

which shows that $\{J_n\}_{n=-1}^{\infty}$ has semi-cycles of length at two. On the other hand, if (15) holds, then we obtain

$$
\begin{aligned}
J_{m+2} &= f(J_{m+1}, J_m) \\
&> f(J_m, J_m) = f(1,1) = J^*
\end{aligned}
$$

and from (12), we find

$$
J_{m+2} = f(J_{m+1}, J_m) < J_{m+1}.
$$

Then,

$$
\begin{aligned}
J_{m+3} &= f(J_{m+2}, J_{m+1}) \\
&< f(J_{m+1}, J_{m+1}) = f(1,1) = J^*,
\end{aligned}
$$

which shows that $\{J_n\}_{n=-1}^{\infty}$ has semi-cycles of length at three.

Next, let (13) holds. Thus, we find

$$
J_1 = f(J_0, J_{-1}) > J_0
$$

and

$$
J_2 = f(J_1, J_0) > J_1
$$

and so on, we get that $\{J_n\}_{n=-1}^{\infty}$ is increasing. Thus, the proof is completed. $\square$

### 2.2. Existence of Periodic Solutions

Next, we obtain necessary and sufficient conditions on $f$ so that every positive solution of Equation (1) is periodic with period two or three.

**Theorem 4.** *Equation (1) has a prime period two solution* $\{J_n\}_{n=-1}^{\infty}$ *where*

$$
J_n = \begin{cases} \rho & \text{if } n \text{ odd} \\ \sigma & \text{if } n \text{ even} \end{cases},
$$

*if and only if*

$$
f(1, \tau) = \tau f(\tau, 1), \tag{16}
$$

*where* $\tau = \rho/\sigma$.

**Proof.** Assume that Equation (1) has a prime period two solution

$$
\ldots, \rho, \sigma, \rho, \sigma, \ldots .
$$

From Equation (1), we have

$$
\rho = f(\sigma, \rho) = f\left(1, \frac{\rho}{\sigma}\right) \text{ and } \sigma = f(\rho, \sigma) = f\left(\frac{\rho}{\sigma}, 1\right).
$$

Thus, we obtain

$$
0 = \rho - \tau\sigma = f(1, \tau) - \tau f(\tau, 1),
$$

where $\tau = \rho/\sigma$.

Next, if we have (16) holds, then we choose, for all $\tau \in \mathbb{R} \setminus \{1\}$

$$
J_{-1} = f(1, \tau) \text{ and } J_0 = f(\tau, 1).
$$

Then, we see that

$$J_1 = f(J_0, J_{-1}) = f(f(\tau, 1), f(1, \tau)).$$

From (16), we find

$$J_1 = f(f(\tau, 1), \tau f(\tau, 1)) = f(1, \tau).$$

Similarly, we can prove that $J_2 = f(1, \tau)$. Hence, it is followed by the induction that

$$J_{2n-1} = f(1, \tau) \text{ and } J_{2n} = f(\tau, 1) \text{ for all } n \geq 0.$$

Therefore, Equation (1) has a prime period two solution, and the proof is complete. □

**Theorem 5.** *Equation (1) has a prime period three solution* $\{J_n\}_{n=-1}^{\infty}$ *where*

$$J_n = \begin{cases} \alpha & \text{for } n = 3r - 1 \\ \beta & \text{for } n = 3r \\ \gamma & \text{for } n = 3r + 1 \end{cases}, \quad r = 0, 1, \ldots$$

*if and only if*

$$\begin{aligned} f(1, s) &= t\, f(s, t); \\ f(t, 1) &= s\, f(s, t), \end{aligned} \tag{17}$$

*where $t = \beta/\alpha$ and $s = \gamma/\alpha$.*

**Proof.** Assume that Equation (1) has a prime period three solution

$$\ldots, \alpha, \beta, \gamma, \alpha, \beta, \gamma, \ldots .$$

From Equation (1), we get

$$\gamma = f(\beta, \alpha), \quad \alpha = f(\gamma, \beta) \text{ and } \beta = f(\alpha, \gamma),$$

and hence,

$$\begin{aligned} \gamma &= f\left(\frac{\beta}{\alpha}, 1\right) = f(t, 1); \\ \alpha &= f\left(1, \frac{\beta}{\gamma}\right) = f\left(1, \frac{t}{s}\right) = f(s, t); \\ \beta &= f\left(1, \frac{\gamma}{\alpha}\right) = f(1, s). \end{aligned}$$

Now, we have

$$t = \frac{\beta}{\alpha} = \frac{f(1, s)}{f(s, t)} \text{ and } s = \frac{\gamma}{\alpha} = \frac{f(t, 1)}{f(s, t)},$$

Hence, we find

$$\begin{aligned} f(1, s) &= t\, f(s, t); \\ f(t, 1) &= s\, f(s, t). \end{aligned}$$

Next, if we have (17) holds, then we choose, for all $t, s \in \mathbb{R} \setminus \{1\}$,

$$J_{-1} = f(s, t) \text{ and } J_0 = f(1, s).$$

Thus, we see that

$$
\begin{aligned}
J_1 &= f(J_0, J_{-1}) = f(f(1,s), f(s,t)) \\
&= f(t\, f(s,t), f(s,t)) = f(t,1).
\end{aligned}
$$

Similarly, we can prove that $J_2 = f(s,t)$ and $J_3 = f(1,s)$. Hence, it is followed by the induction that

$$
J_{3n-1} = f(s,t), \ J_{3n} = f(1,s) \text{ and } J_{3n+1} = f(t,1) \text{ for all } n \geq 0.
$$

Therefore, Equation (1) has a prime period three solution, and the proof is complete. □

## 3. Discussion and Examples

**Example 1.** *Consider the difference equation*

$$
J_{n+1} = a + \frac{J_{n-1}}{J_n}, \tag{18}
$$

*where a is a real number. Note that*

$$
f(u,v) = a + \frac{v}{u}
$$

*and*

$$
\mu = \frac{-1}{a+1}
$$

*From Theorem 1, if $\mu > 0$ (when $a < -1$), then Equation (18) is locally asymptotically stable if and only if $a < -2$. In addition, let $\mu < 0$ (when $a > -1$), we get that Equation (18) is locally asymptotically stable if $a > 1$ and unstable if $-1 < a < 1$. Thus, we can assert that Equation (18) is locally asymptotically stable if $a \in (-\infty, -2) \cup (1, \infty)$ and unstable if $a \in (-2, 1)$.*

**Remark 2.** *Equation (18) was studied by Amleh et al. in [21] and Hamza in [22] when $a \geq 0$ and $a < 0$, respectively. Our results in Example 1 are consistent with the results in [21,22].*

**Example 2.** *Consider the difference equation*

$$
J_{n+1} = \alpha + \beta \frac{J_n}{J_{n-1}} + \gamma \frac{J_{n-1}}{J_n}, \tag{19}
$$

*where $\alpha, \beta$ and $\gamma$ are positive real numbers. Note that*

$$
f(u,v) = \alpha + \beta \frac{u}{v} + \gamma \frac{v}{u}
$$

*and*

$$
\mu = \frac{\beta - \gamma}{\alpha + \beta + \gamma}.
$$

*By using Theorems 1, we see that Equation (19) is locally asymptotically stable if $\beta > \gamma$ or $\beta < \gamma < \alpha + 3\beta$ and unstable if $\gamma > \alpha + 3\beta$.*

**Remark 3.** *Equation (19) was studied by Elsayed in [23]. He showed that Equation (19) is locally asymptotically stable if*

$$
2|\beta - \gamma| < \alpha + \beta + \gamma. \tag{20}
$$

*From our results in Example 2, if we choose $\alpha = 0.1$, $\beta = 2$ and $\gamma = 0.1$, then Equation (19) is locally asymptotically stable see (Figure 1). However, the values of $\alpha, \beta$ and $\gamma$ do not fulfill the condition (20). Thus, our results here extend and improve results of Elsayed [23].*

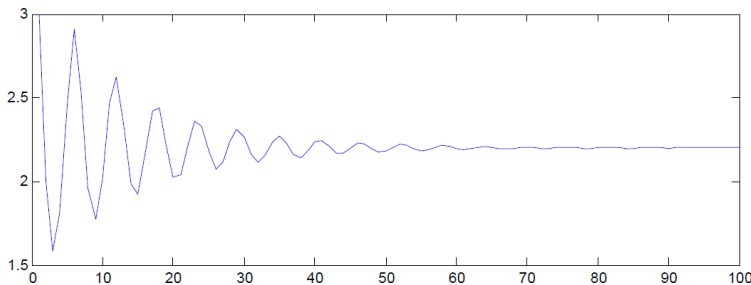

**Figure 1.** The stable solution corresponding to differences Equation (19).

**Remark 4.** *By using Theorem 1, we can state new necessary and sufficient conditions for locally asymptotically stability of several equations that previously have been studied, including [13,23–25], etc.*

**Corollary 1.** *Equation*

$$J_{n+1} = a + b \left( \frac{J_n}{J_{n-1}} \right)^{\alpha},$$

(21)

*where $a, b$ and $\alpha$ are real numbers, has no prime period two solution if $a, b$ and $\alpha$ are positive. In addition, Equation (21) has a prime period two solution if and only if*

(**i**) $a/b < -(2\alpha + 1)$, $\alpha > 0$ *and* $J_{-1}J_0 > 0$;

(**ii**) $a/b > 1$, $\alpha$ *odd and* $J_{-1}J_0 < 0$;

(**iii**) $a/b < -1$, $\alpha$ *even and* $J_{-1}J_0 < 0$.

**Proof.** From Theorem 4, we have that Equation (21) has a prime period two solution $\ldots, \rho, \sigma, \rho, \sigma, \ldots$ , if and only if

$$\frac{a}{b} = \left( \frac{\tau^{2\alpha+1} - 1}{\tau^{\alpha}(1 - \tau)} \right).$$

(22)

We note that $\frac{\tau^{2\alpha+1}-1}{\tau^{\alpha}(1-\tau)} < 0$ for all $\tau \in \mathbb{R}^+$. Then Equation (21) has no prime period two if $a$ and $b$ are positive. Now, we define

$$H(\tau) = \frac{\tau^{2\alpha+1} - 1}{\tau^{\alpha}(1 - \tau)}.$$

Hence, we see that

$$H(\tau) < -(2\alpha + 1), \text{ for } \tau \in \mathbb{R}^+ \setminus \{1\} \text{ and } \alpha > 0.$$

(23)

Thus, we get

$$\frac{a}{b} < -(2\alpha + 1).$$

On the other hand, if $\tau \in \mathbb{R}^-$, then we find $H(\tau) < -1$ if $\alpha$ odd and $H(\tau) > 1$ if $\alpha$ even. Hence, the proof is complete. $\square$

**Example 3.** *If we take*

$$a = -7/2, b = 1, \alpha = 1, J_{-1} = -3 \text{ and } J_0 = -3/2 \text{ (see Figure 2a)};$$

$$\text{or } a = 43, b = 8, \alpha = 3, J_{-1} = -21 \text{ and } J_0 = 42 \quad \text{(see Figure 2b)};$$

$$\text{or } a = 61, b = -9, \alpha = 2, J_{-1} = 60 \text{ and } J_0 = -20 \quad \text{(see Figure 2c)},$$

*then Equation (1) has prime period two solutions.*

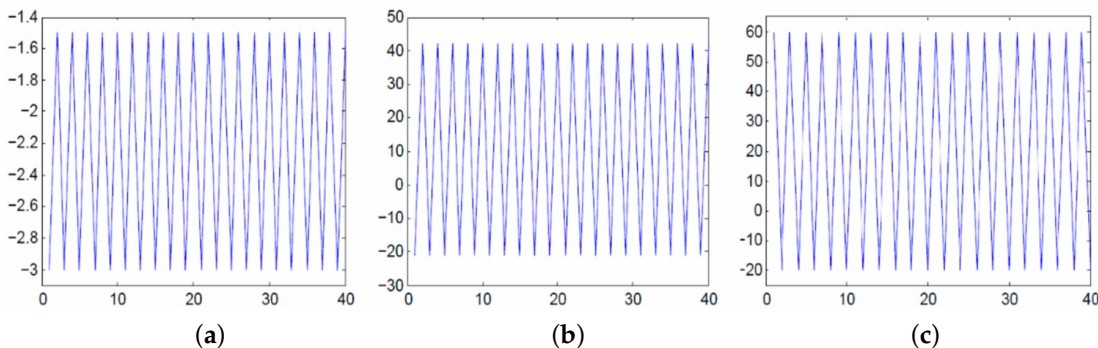

**Figure 2.** Prime period two solutions of Equation (21).

**Remark 5.** *Stevic [17] [in Theorem 4.1] studied the periodicity of the difference Equation (21) when $b = 1$ and $a, \alpha \in (0, \infty)$, and proved that there are no positive prime two-periodic solutions of (21). Thus, our results in Theorem 1 extend and complement the results of Stevic [17].*

**Corollary 2.** *Equation (21) has a periodic solution of prime period three if and only if*

$$t \left( \frac{s^{2\alpha} - t^{\alpha-1}}{s^{\alpha+1} - t^{2\alpha}} \right) = s^\alpha \left( \frac{1 - t}{1 - s} \right)$$

*and*

$$a = b \left( \frac{s^{\alpha+1} - t^{2\alpha}}{t^\alpha (1 - s)} \right),$$

*where $t = J_0 / J_{-1}$ and $s = J_1 / J_{-1}$.*

The proof is immediate (from the proof of Theorem 5) and hence is omitted.

**Example 4.** *If we take $a = -0.6722, b = 1, \alpha = 1, J_{-1} = 0.9518$ and $J_0 = -1.1722$, then Equation (21) has a prime period three solution (see Figure 3).*

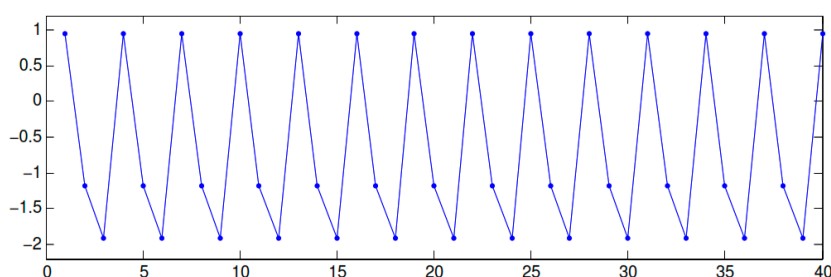

**Figure 3.** Prime period three solutions of Equation (21).

**Remark 6.** *By using our results, we can study the qualitative behavior of many difference equations, for example*

$$x_{n+1} = a + b e^{-x_{n-1}/x_n},$$
$$x_{n+1} = \sqrt{\frac{a x_{n-1}}{b x_n + c x_{n-1}}}.$$

*In addition, by the change of variables $x_n = \frac{b}{J_{n+1}} - a$, the equation*

$$x_{n+1} = \frac{a + x_n}{a + x_{n-1}}$$

*reduces to the equation*

$$J_{n+1} = \frac{bJ_n}{aJ_n + J_{n-1}}$$

*which is a special case from Equation (1).*

## 4. Conclusions

In this paper, we study a general class of difference equations (covering many equations studied by other authors or that have never been studied before), as a means of establishing general theorems, for the asymptotic behavior of its solutions. Here, we state new necessary and sufficient conditions for local asymptotic stability of Equation (1). In addition, by using new technique in Theorems 4 and 5, we establish new necessary and sufficient conditions and these conditions are very useful in studying the existence of periodic solutions of period two and three. Our results here extend and complement many recent ones in the literature. Suitable illustrative examples have also been provided.

**Author Contributions:** The authors have made the same contribution. All authors read and approved the final manuscript.

**Funding:** This research received no external funding.

**Conflicts of Interest:** The authors declare no conflicts of interest.

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
