# Peer review of "Some Qualitative Behavior of Solutions of General Class of Difference Equations"

_mathematics, doi:10.3390/math7070585_

Round 1

Reviewer 1 Report

\documentclass{amsart}

\begin{document}

\centerline{\it Referee's report on Manuscript ID: mathematics-510670:}\medskip

\centerline{\it Some

qualitative behavior of solutions of general class of difference equations}\medskip

In this paper the authors studied the difference equation

\[

x_{n+1}=f(x_n,x_{n-1}),\ n=0,1,...

\]

where $f$ is homogenous of degree zero    and       continuous function. More precisely they studied the asymptotic stability of the equilibrium, the semi-cycle analysis and the periodicity of the above equation. Finally they gave examples to illustrate their results.  Since difference equations have many applications in the applied sciences the paper could be accepted for publication.\par

I have two comments:\\

1. In my opinion Theorem 2.1 and Remark 2.1 follows immediately from Theorem 1.3.4 of the book "Global behavior of nonlinear difference equations of higher order with Applications", Kluwer Academic Publishers.\\

2. The authors found necessary and sufficient conditions for the existence of periodic solutions of period two and three. What about for period greater than three?

Some corrections  must me done.\\

1. $P6^{4}.$   "add" should be "odd".

2. $P6^{15}.$ the authors should set spaces in "wehave(2.12),thenwechoose...."

3. $P7^{17}.$ "$f(y_0,y_1)$"  should be "$f(y_0,y_{-1})$".

\end{document}

Author Response

Pl. find the attached file for clarifications of  all three referee reports.

 Thanks,

Dimplekumar N Chalishajar

Reviewer 2 Report

report in PDF file

Author Response

(The authors gave the same response as above.)

Reviewer 3 Report

in attached pdf file

Author Response

(The authors gave the same response as above.)

Round 2

Reviewer 1 Report

In my opinion the paper after the corrections  can be accepted for publication.